# Exercise Training in Elderly Cancer Patients: A Systematic Review

**DOI:** 10.3390/cancers15061671

**Published:** 2023-03-08

**Authors:** Francesco Giallauria, Crescenzo Testa, Gianluigi Cuomo, Anna Di Lorenzo, Elio Venturini, Fulvio Lauretani, Marcello Giuseppe Maggio, Gabriella Iannuzzo, Carlo Vigorito

**Affiliations:** 1Department of Translational Medical Sciences, University of Naples Federico II, via S. Pansini 5, 80131 Naples, Italy; 2Faculty of Sciences and Technology, University of New England, Armidale, NSW 2351, Australia; 3Geriatric Clinic Unit, Geriatric-Rehabilitation Department, University Hospital, 43126 Parma, Italy; 4Cardiac Rehabilitation Unit and Department of Cardiology, Azienda USL Toscana Nord-Ovest, “Cecina Civil Hospital”, 57023 Cecina, Italy; 5Cognitive and Motor Center, Medicine and Geriatric-Rehabilitation Department of Parma, University Hospital of Parma, 43126 Parma, Italy; 6Department of Clinical Medicine and Surgery, University of Naples Federico II, via S. Pansini 5, 80131 Naples, Italy

**Keywords:** cancer, exercise training, cardio-oncology, elderly, cardiotoxicity, frailty, quality of life, biology of exercise, rehabilitation, prehabilitation, cardiac rehabilitation

## Abstract

**Simple Summary:**

The incidence and prevalence of cancer mainly affect the geriatric population. In this segment of the population, streamlining cancer care and making it more sustainable is essential. The objective in the care of the elderly cancer patient is not simply the control of the disease but above all the quality of life and the prevention of disability. With this systematic review of the literature, we want to analyze the state of the art of oncological care in the elderly patient by proposing physical exercise both before and after specifically oncological treatments as a driving force of good clinical practice.

**Abstract:**

Due to the aging of the population, in 70% of cases, a new cancer diagnosis equals a cancer diagnosis in a geriatric patient. In this population, beyond the concept of mortality and morbidity, functional capacity, disability, and quality of life remain crucial. In fact, when the functional status is preserved, the pathogenetic curve towards disability will stop or even regress. The present systematic review investigated the effectiveness of physical exercise, as part of a holistic assessment of the patient, for preventing disability and improving the patient’s quality of life, and partially reducing all-cause mortality. This evidence must point towards decentralization of care by implementing the development of rehabilitation programs for elderly cancer patients either before or after anti-cancer therapy.

## 1. Introduction

Advanced age is one of the major risk factors that predispose to the development of chronic diseases. The enormous technological and therapeutic development has led to the progressive increase in the prevalence of many chronic diseases, which just over 15 years ago were considered fatal in the short term. The oncological disease is the one in which these therapeutic advances have had the most impact; in perspective, more innovative therapeutic aids will likely prolong the survival or even well-manage the course of patients with advanced stage oncological disease [1]. For several years now, the increase of older multimorbid patients with associated clinical complexity has required more attention towards stratification methods to improve personalized treatment.

Given the higher number of older patients, the comprehensive geriatric assessment seems the best approach [2].

A multidomain care model approaching the geriatric patient is the best strategy to move beyond the survival and disease model and target the quality of life too. Physical exercise is an important component of multidomain strategies of oncological care adoptable in every stage of cancer disease. 

Taking into consideration the epidemiological trends [3], the focus on elderly patients in this setting is mandatory. The elderly patient is unique as such the comprehensive geriatric evaluation also certifies the patient’s social condition which is essential for adapting the therapeutic approach. The biopsychosocial status of the elderly patient must be preserved, and multicomponent interventions have already demonstrated their effectiveness in arresting the clinical trend towards disability [4]. Rehabilitation or oncological prehabilitation in the elderly patient would have the peculiarity of directing the patient towards integrated pathways.

The aim of this approach will likely improve the prognosis of the disease but above all will provide “multidimensional support” to the elderly oncological patient and provides a social network known for effectively slowing the decline towards disability [4,5]. The skills and toolkit that have proved to be effective over the years in setting up cardiological rehabilitation pathways in elderly patients [6] apply also to cancer. The tools of the comprehensive geriatric assessment, combined with the classification of the patient’s functional capacity through the cardiopulmonary test, will be able to direct clinicians towards an adequate stratification of patients to set up rehabilitation or prehabilitation pathways tailored to the patient’s needs.

The main difference between the oncological rehabilitative pathways in elderly patients compared to the traditional cardiology rehabilitative pathways would be precisely the integration of the oncological rehabilitative pathway into the diagnostic-therapeutic process of the oncological pathology. While rehabilitative cardiology is a fundamental tool of cardiac care at the end of the diagnostic-therapeutic path, rehabilitative oncology is an integrated path of oncological treatments to be customized according to the patient’s preference. In this field, rehabilitation is part of treatment and is located not only at the end of the diagnostic-therapeutic process but also in the intermediate and initial phases of the prehabilitation programs.

Rehabilitative cardiologists were among the first to realize the potential of physical exercise as a therapeutic tool in cancer patients. Traditionally, the first meeting point between oncological disease and rehabilitative cardiology has been in the treatment of cardiac dysfunction induced by many anticancer treatments.

In 2016, the guidelines of the American Society of Clinical Oncology suggested that, in certain patients, physical exercise could be used as strategies for preventing and treating cardiac disease induced by cardiotoxic drugs, such as anthracyclines, trastuzumab, or radiotherapy [7].

There is numerous evidence in the literature to support a systemic beneficial effect of exercise both in cardiovascular [8] and in oncological settings [9], regardless of the phase of the disease. Cardiovascular and oncological pathology share several risk factors which not only feed themselves in a vicious circle but are also positively influenced by physical exercise [10]. 

Cardiovascular mortality is the one that most begins to influence long-term survival in cancer patients, especially due to the rapid aging of the population [11]. In long-term cancer survivors, cardiovascular mortality increased from 1.3 to 3.6 times [12] while the incidence of cardiovascular risk factors increased from 1.7 to 18.5 times compared to control without cancer [13]. Considering this evidence, the birth of cardio-oncological rehabilitation naturally arose, where the professional skills and structures of cardiac rehabilitation units were made available to oncological patients [14]. The clinical need then turned into a real therapeutic algorithm (CORE, Cardio-Oncology REhabilitation) postulated by the American Heart Association and endorsed by the American Cancer Society [15]. In the elderly population, other aspects should be carefully considered: frailty, social isolation, malnutrition, cognitive decline, dementia, disability, other geriatric syndromes, and above all the quality of life. Concisely, the complexity of the geriatric cancer patient requires a mandatory multidomain approach in which, when possible, the effects of physical exercise can mitigate or even reverse the disease states. 

This systematic review focused on the effects of exercise on older cancer patients. Much more remains to be conducted on the correct stratification of patients and in the implementation of the settings necessary to carry out rehabilitation or even prehabilitation cycles in cancer patients. However, the awareness that in the elderly cancer patient, physical training is a fundamental clinical aid that modern medicine cannot ignore is essential.

## 2. Elderly Population: When an Epidemiological Shift Turns into a New Model of Care

The most recent epidemiological trends indicate that thanks to earlier diagnosis and more effective treatments, the prevalence of long-term cancer patients is increasing year on year [16]. These same trends, in addition, indicate that approximately 30% of new cancer diagnoses (regardless of histotype) are made after age 70 and, for certain histotypes and disease stages, the 5-year survival is over 85% [16]. The dynamism of epidemiological trends is one-way: in 2040 in the United States the prevalence of long-term cancer patients is estimated at 26.1 million individuals, 73% of these will be 65 or older [3]. Thus, in most cases, talking about a long-term cancer survivor means talking about a geriatric patient, with all the consequences that this definition brings.

The first consequence of this epidemiological situation is that cancer patients have several comorbidities. The most frequent are heart failure, chronic obstructive pulmonary disease, and diabetes mellitus [3]. 

Identification and proper treatment of these comorbidities before cancer diagnosis affects the outcome when initiating anti-cancer treatment [17]. The presence of comorbidities negatively impacts the performance status of these patients [18]. 

Therefore, the need arises for a holistic evaluation of the patient and post-treatment rehabilitation programs to maintain a sufficient functional degree of the patients [19,20]. This initial analysis has profound implications: the traditional model of care where the follow-up is all oriented according to the behavior of the disease (relapse or remission) is no longer sufficient. It is evident that the oncologist while maintaining his priority role, must be integrated into a multidisciplinary system that takes charge of not only the oncological disease but the elderly cancer patient. 

The American National Research Council was among the first institutions to indicate the essential points of care for long-term cancer survivors. The reference to an innovative, multidisciplinary, person-centered treatment model is explicit in four fundamental points: (1) prevention of relapses, new cases, and the long-term effects of anti-cancer treatments; (2) active surveillance of both oncological disease and its medical and psychosocial sequelae; (3) management of the consequences of treatments, including social and psychological consequences; (4) coordination between different health professionals to deal with all the needs of cancer survivors [21].

Several innovative care models have been proposed over the years as an evolution of the traditional model. In particular, the most studied in clinical trials were general practitioner-led care; care shared between the general practitioner and cancer specialist; and oncology nurse-led care. These models all go in the direction of decentralizing the follow-up of long-term cancer survivors by promoting a system that in most cases is just as effective but less expensive and above all more tolerable for the patient. For example, patients who receive a follow-up by a general practitioner are more satisfied, with a lower expense for the health system than the follow-up by specialists [22]. Several meta-analyses have shown that a general practitioner-led care model is not inferior to the traditional model in the diagnosis of disease recurrence [23,24,25,26,27,28,29,30]. This model, as well as others, contributes to empowering the patient about his/her health. Several meta-analyses have shown an improvement in the quality of life in patients who were supported to have a healthier lifestyle, especially through the promotion of exercise [31,32].

In another model of care, where follow-up is shared between the general practitioner and the oncologist, clinical outcomes were obtained [33] at a lower cost to the health system and with greater patient satisfaction [26,30,34]. Finally, a last alternative model has been hypothesized and several studies indicate that it is applicable in a wide range of clinical situations: a follow-up entrusted to nurses with specific training in the oncology field [33]. Many clinical trials have been performed, in different disease settings in which patients with different types of cancer have been included. 

Overall, the evidence has shown that this follow-up model is not inferior to the standard model in the detection of disease recurrence [24,25,26,27,35,36,37]. Notably, in this follow-up model, a lower cost for the health system was highlighted [38].

It is evident that all these models presented are moving towards decentralization of care and towards a holistic approach to the patient with cancer. Therefore, multidimensional oncological rehabilitation is born, in which the purely oncological aspects of the disease are integrated with the biopsychosocial aspects of the person. This approach has shown considerable effectiveness in improving the quality of life of patients, the physical and psychological aspects of the disease, and even the return to work [39,40].

One of the cornerstones of multidimensional cancer rehabilitation is physical exercise which has been shown to improve physical capacity and fatigue typical of cancer patients [41]. Improvements were also highlighted regarding the mood of the patients [41]. Finally, the effect on the improvement of survival of cancer patients who perform physical exercise should be remarked [42]. The follow-up models analyzed here are particularly valid in an increasingly elderly population where a long-term cancer survivor is synonymous with a geriatric patient. 

Decentralized and multidimensional approaches are particularly necessary in this population where the follow-up of the disease must be a moment of sharing between different specialists and health professionals to guarantee a therapeutic approach that is most tailored to the patient’s needs.

## 3. Exercise and Cancer: Pathophysiological Basis 

Exercise has beneficial effects on all-cancer pathogenetic hallmarks [43] (Figure 1). Weight-loss associated with physical activity causes the reduction in circulating levels of IGF-1 and the associated mitogenic signal regulated by the cascade of MAP kinases [44,45]. In healthy men in their 60s, exercise reduces circulating IGF-1 levels and increases p53 expression in prostate cancer cells when stimulated with serum from this group of men [46]. 

One of the most important mechanisms of carcinogenesis is the evasion of cell replication suppression signals. There is numerous evidence indicating that increased levels of physical exercise have effects on the pathways that regulate cell growth processes such as p21 expression, PTEN, m-TOR and retinoblastoma signaling [47,48,49,50]. Apoptosis is another target that cancer cells try to escape.

The effects of physical exercise have been shown to be effective in re-establishing correct apoptotic mechanisms through the downregulation of the antiapoptotic protein BCL-2 [49,51,52]. The effectors of the apoptotic cascade are the caspases; exercise has been shown to be effective in increasing caspase activity [53,54]. Exercise is also capable of increasing the activity of Bax and Bak, two well-known proapoptotic proteins [55]. The effects of physical exercise on the tumor microenvironment are very important. Exercise has been shown to make the tumor microenvironment “more physiological”, for example by improving tissue perfusion of mouse models of human breast and prostate tumors [56,57,58,59]. 

Some authors have hypothesized that better tumor perfusion makes the tumor tissues more sensitive to chemotherapy drugs [59]. Interestingly, a study has shown that the combination of exercise and cyclophosphamide reduced the growth of mouse breast cancer compared to a cycle of cyclophosphamide alone [53].

The beneficial effects of exercise on a proinflammatory state, a known pathogenetic driver of carcinogenesis, are well-documented [8]. At the tissue level, the effects of physical exercise modify the tumor microenvironment through a reduction of tumor-infiltrating macrophages [60,61,62]. Furthermore, physical exercise promotes the formation of M1 macrophages which, unlike M2, have a th1-type cytokine expression, with an antitumor effect [63,64]. Interestingly, physical exercise does not act directly but indirectly on some cytotypes of the immune system, such as NK cells which have an important immunological surveillance function and whose function is strongly compromised at the level of tumor tissues. One study has shown that exercise makes the tumor microenvironment more prone to infiltration by NK cells, through increased expression of specific ligands [65,66]. Although many of these findings have been observed in preclinical studies, there is also increasing evidence in clinical trials. The main effector cell of the regulation of immunity is Treg, as fundamental in the prevention of self-administration disorders as it is deleterious if dysregulated in the tumor microenvironment [67]. Exercise has been inversely associated with the expression of circulating t-regs compared to sedentary women [68]. Some authors begin to hypothesize that a lower expression of t reg at the tissue level can positively influence the response to the new immunotherapeutic drugs [67,68]. 

Finally, several pieces of evidence indicate that the muscle can be classified as a real endocrine organ, and, consequently, more integrated into a much more complex system. The effectors of these endocrine functions are circulating molecules produced by the muscle, the myokines. Myokines act systemically as anti-inflammatory factors, have an insulin-sensitizing action, and promote thermogenesis at the level of adipose tissue cells [69]. Some myokines are capable of inducing apoptosis in some tumors, mainly breast [53], and colon cancers [70]. Although the preclinical evidence can give an important insight into the importance of physical exercise in the patient with neoplastic disease, it is essential to confirm this evidence also in randomized controlled clinical trials. A recent randomized controlled study of 318 breast cancer survivors showed that physical exercise reduces inflammatory mediators (such as C-reactive protein) that are associated with disease recurrence and cardiovascular death [71].

It is evident that the positive effects of physical exercise are appreciated both at the molecular and clinical levels in cancer patients. However, in the heterogeneous melting pot of numerous facets, which is the geriatric older patient, particular attention should be devoted to the depressive syndrome which even negatively impacts the survival of patients [72]. In elderly patients, this association between cancer and depression is even more important. In fact, social isolation, and loss of functional capacity, makes geriatric patient particularly prone to developing mood disorders.

All these triggers, in an older patient with cancer and accelerated frailty, can lead to a speeding up of the neurodegenerative processes that cause frank dementia [5]. Furthermore, in this case, physical exercise, perhaps through the involvement of elderly patients in community centers where it is possible to establish social relationships, has proved useful in the treatment of depressive symptoms [73]. All mentioned studies are summarized in Table 1. 

## 4. Exercise in Older Cancer Patients 

In the previous section, the pathophysiological foundations of the utility of physical exercise in the oncological setting were laid. To do this, numerous studies have been reported that indicate that physical exercise can intercept or even stop the pathophysiological processes that underlie the hallmark of the neoplastic transformation of somatic cells. It is well known that less than a third of elderly patients eligible to be part of randomized controlled trials in an oncological setting are not recruited due to inclusion criteria that do not reflect real-life and social or welfare issues [74]. 

The aim of this systematic review of the literature is to highlight the efficacy of physical exercise in preserving the functional capacity of elderly cancer patients. Translating how clinical trial patients are recruited into a geriatric setting can be challenging. The elderly patient has specific peculiarities, and their correct classification is essential to proportionate the type of diagnostic therapeutic path to be undertaken. Terms such as “frailty” and “disability” are just some of the ways that the scientific community must stratify the functional autonomy of an elderly patient and very often there is no agreement between the different researchers on the cut-offs for including a patient in one category rather than the other. There is more agreement, however, in defining the cut-off of 65 years of age to define a geriatric patient, and only in recent years has there been agreement in identifying in the older cancer survivor a patient diagnosed with oncological pathology who is over 65 years old. Clinical studies designed to identify the effect of a physical exercise specifically in this age range are very few now. The first study that demonstrated the effectiveness of physical training in this clinical setting showed, in addition to the improvement in the patients’ quality of life, also an increase in the Short Physical Performance Battery (SPPB), a parameter of enormous importance in the geriatric field [75,76].

For this reason, we selected randomized controlled trials on patients whose mean age is 65 years or older and which were conducted within the last 20 years. Before the 2000s there was very little evidence of the usefulness of physical exercise in the elderly patient with cancer but above all, there was no chronicitazion of the neoplastic disease that we have been observing in recent years and will observe in the future [3]. Most of the evidence comes after 2010 when the American Society of Sports Medicine hoped for a therapeutic path for patients with oncological pathology that included physical exercise among the various actors [77].

Two researchers (F.G. and C.T.) independently screened the main search engines (MEDLINE, EMBASE, and SCOPUS) with all possible combinations of the following keywords: “exercise, training, cancer, patient, survivor”. A total of 11,216 records were identified. PRISMA diagram is reported in Figure 2. The included studies [76,78,79,80,81,82,83,84,85,86,87,88,89,90,91,92,93,94] are shown in Table 2. The risk of bias of the included studies is shown in Figure 3.

The systematic review followed the recommendations of the Preferred Reporting Items for Systematic Reviews and Meta-Analyses (PRISMA). The protocol has not been registered.

Over the years and with the increase in scientific knowledge in the onco-rehabilitative field, the validity of physical exercise in cancer patients and the onco-geriatric field has become irrefutable. The enormous attention of the scientific community is certainly due to the epidemiological shift that the population of cancer patients will have to face in the coming years. There is no precise correspondence between the type of physical exercise to be performed and the tumor histotype. Moreover, the evidence is still very limited on this subject and the interventions range from simple aerobic exercise to resistance-type exercise to HIIT programs. What is certain is that exercise is effective in all stages of the disease and that training programs, in most cases, do not interfere with concomitant anticancer treatments [95], finally a greater muscle strength of the lower limbs allows better stability during walking, an essential prerogative to avoid falls and the consequent functional impairment [96].

To Feb 2023, about 36 clinical trials are currently underway aiming at investigating the effects of exercise in cancer patients over the age of 65. Studies are beginning to evaluate the effects of exercise-based interventions on minorities in the population as well [97,98]. It should be emphasized that initial and very recent evidence indicates that physical exercise can also have positive effects on disease relapses [99]: among patients with stage III colon cancer enrolled in a trial of postoperative treatment, larger volumes of physical exercise correlated with better disease-free survival. 

The SARS-CoV-2 pandemic, with the consequent difficulties due to social distancing and the lockdown, has pushed medicine, when possible, towards a more telematic direction. The cancer rehabilitation settings went in the same direction, increasing the range of action by proposing multimodal follow-ups where in addition to promoting physical exercise, behavioral measures were also proposed, for example regarding nutrition. These telerehabilitation measures have not only proved effective, tolerable, and applicable even in elderly patients who have long survived cancer, but in some cases, they have also demonstrated the prevention of functional autonomy decline, a data of extreme importance in the modern medical panorama [84,100,101,102,103].

The literature provides evidence for the notion that telemedicine in the oncological setting is effective and tolerable, even in elderly patients. The remote consultation enjoyed a renewed lifeblood during the pandemic period where it proved to be essential during quarantine. The results are effective and valid in different disease settings and different tumor histotypes [104,105,106,107,108,109,110]. These findings confirm the data of the RENEW study, a randomized and controlled clinical trial that in 2009 showed that telephone consultation combined with the sending of informative material via e-mail, with the aim of improving healthy behaviors, reduced functional decline in long-term survivors of breast, prostate, and colorectal cancer between the ages of 65 and 91 [111]. In the same population, the higher the adherence to telephone consultations, the better the outcomes on exercise tolerance, functional capacity, mood, and body composition [112].

Prospectively, telemedicine approaches are effective and safe in elderly cancer patients with the increasing contribution of artificial intelligence [113,114]. However, more attention should be paid to the sociability and social networks of patients, particularly in the last decades of life and in independent patients. Stronger social networks have been shown as valid strategies to delay the loss of autonomy [4,5]. Physical exercise, in this setting, has been shown to be effective in improving mood disorders, anxiety, and sociability in elderly patients suffering from cancer [78,79,80]. The caregiver of cancer patients can be particularly burdened with worries, anxieties, and stress that very often accompany the diagnosis and treatment of cancer. One study showed that exercise in couples coping with cancer improved both physical and cognitive parameters [115]. The geriatric relevance of these initial findings is very important as improved sociability and a reduction in mood disorders contribute to the prevention of disability.

**Table 2 cancers-15-01671-t002:** Main characteristics and results of randomized controlled trials evaluating oncological rehabilitation in the elderly patient.

Authors, Year.	Cancer Type	Mean Age	Follow-Up	Intervention and Methods	Main Results
Focht et al., 2019, [78]	Prostate	66	3 months	Single-blind, 2-arm, randomized controlled Individualized trial, 32 prostate cancer patients undergoing androgen deprivation therapy were randomly assigned to a 12-week of group-mediated cognitive behavioral exercise and dietary intervention (n = 16) or standard care treatment (n = 16)	Lifestyle intervention yielded more favorable improvements in relevant social cognitive outcomes relative to standard care
Galvão et al., 2021, [79]	Prostate	69	12 months	A total of 135 prostate cancer patients aged 43–90 years on androgen deprivation therapy were randomized to twice weekly supervised impact loading and resistance exercise, supervised aerobic and resistance exercise, and usual care	Various supervised exercise modes (aerobic, resistance, and impact loading are effective in reducing psychological distress in men with prostate cancer.
Poh Loh et al., 2019, [80]	Solid cancer	68	6 weeks	Exploratory secondary analysis of a randomized controlled trial. Patients were randomized to exercise (home-based, low-to-moderate intensity progressive walking and resistance training program) or usual care (control) for the first six weeks of chemotherapy.	Among older cancer patients receiving chemotherapy, a 6-week structured exercise program improved anxiety and mood, especially among those participants with worse baseline symptoms
Winters-Stone et al., 2016, [115]	Prostate	70	6 months	Single-blind randomized controlled trial comparing exercise together (patients and his/her spouse, 32 couples) to usual care (32 couples). The exercise consisted of a progressive strength training program.	Men exercising together became stronger in the upper body (*p* < 0.01) and more physically active (*p* < 0.01) than usual care. Women in Exercising Together increased muscle mass (*p* = 0.05) and improved upper (*p* < 0.01) and lower body (*p* < 0.01) strength and physical performance battery scores (*p* = 0.01) more than usual care
Winters-Stone et al., 2021, [76]	Breast	72	18 months	Early-stage, post-treatment, older (≥65 years) breast cancer survivors (n = 114) were randomized to 12 months of supervised aerobic (n = 37), resistance (n = 39), or stretching (active control; n = 38) training followed by 6 months of unsupervised home-based training	Supervised exercise can improve strength and physical functioning among older breast cancer survivors. Resistance training may lead to better improvements compared to aerobic or flexibility training, whether in a supervised or unsupervised setting
Galvão et al., 2017, [81]	Prostate with bone metastases	70	3 months	A total of 57 prostate cancer patients with bone metastases (pelvis, 75.4%; femur, 40.4%; rib/thoracic spine, 66.7%; lumbar spine, 43.9%; humerus, 24.6%; other sites, 70.2%) were randomized to multimodal supervised aerobic, resistance, and flexibility exercises undertaken thrice weekly (n = 28) or usual care (n = 29)	Multimodal modular exercise in prostate cancer patients with bone metastases led to self-reported improvements in physical function and objectively measured lower body muscle strength with no skeletal complications or increased bone pain
Scott et al., 2021, [82]	Lung	65	17 weeks	A total of 90 lung cancer survivors with poor cardiorespiratory fitness were randomly allocated to receive 48 consecutive supervised sessions thrice weekly of Aerobic training [27], Resistance Training [26], Combination training [23], Stretching control [26]	Aerobic training and combination training significantly improved VO_2_ peak in lung cancer survivors
Zimmer et al., 2017, [83]	Colorectal	68	4 weeks	Thirty patients (stage IV) undergoing outpatient palliative treatment were randomly assigned to an intervention or control group (IG, n = 17; CG, n = 13). The IG participated in an eight-week supervised exercise program including endurance, resistance, and balance training (2×/week for 60 min) whereas the CG received written standard recommendations to obtain physical fitness.	Positive effects of a multimodal exercise program on chemotherapy-induced peripheral neuropathy, balance, and strength on patients with colorectal cancer in a palliative setting, thereby consequently increasing patients’ quality of life.
Mikkelsen et al., 2022, [84]	Advanced pancreatic, biliary tract, or non-small cell-lung cancer	72 (median)	12 weeks	Eighty-four older adults (≥65 years) with advanced pancreatic, biliary tract, or non-small cell lung cancer who received systemic oncological treatment were randomized 1:1 to an intervention group or a control group. The intervention was a 12-week multimodal exercise-based program including supervised exercise twice weekly followed by a protein supplement, a home-based walking program, and nurse-led support and counseling	A 12-week multimodal exercise intervention with targeted support proved effective in improving physical function in older patients with advanced cancer during oncological treatment
Owusu et al., 2022, [85]	Breast	72	1 year	Randomized controlled trial. The interventions included 20 weeks of supervised moderate-intensity aerobic and resistance training followed by 32 weeks of unsupervised exercise (n = 108) and a 20-week support group program followed by 32 weeks of unsupervised activity (n = 105)	Combined aerobic and resistance exercise appears to improve physical performance in older breast cancer survivor
Mardani et al., 2020, [86]	Prostate	69	12 weeks	80 patients were randomly allocated to intervention (exercise program, n = 40) and control (n = 40) groups in a single-blind, parallel, randomized controlled trial	In the intervention group, statistically significant improvements in physical, emotional, social, and sexual function were reported. Moreover, the patients in this group reported reduced fatigue, insomnia, constipation, diarrhea, urinary, bowel, and hormonal treatment-related symptoms in comparison with before the exercise program
Taaffe et al., 2017, [87]	Prostate	69	1 year	163 prostate cancer patients aged on androgen deprivation therapy were randomized to exercise targeting the musculoskeletal system (impact loading + resistance training; n = 58), the cardiovascular and muscular systems (aerobic + resistance training; n = 54), or to usual care/delayed exercise (n = 51)	Different exercise modes have comparable effects on reducing fatigue and enhancing vitality during ADT. Patients with the highest levels of fatigue and lowest vitality had the greatest benefits
Wall et al., 2017 [88]	Prostate	69	6 months	Ninety-seven men with localized prostate cancer receiving ADT were randomized to either exercise (n = 50) or usual care (n = 47). The supervised exercise was undertaken twice weekly at moderate to high intensity.	Combined aerobic and resistance exercise program has a significant favorable effect on cardiorespiratory capacity, resting fat oxidation, glucose, and body composition despite the adverse effects of hormone suppression.
Gaskin et al., 2016, [89]	Prostate	67	12 weeks	Secondary analysis on data from a multicentre cluster randomized controlled trial in which 15 clinicians were randomly assigned to refer eligible patients to an exercise training intervention (n = 8) or to provide usual care (n = 7). Data from 119 patients (intervention n = 53, control n = 66) were available for this analysis	Men with prostate cancer who act upon clinician referrals to community-based exercise training programs can improve their strength, physical functioning, and, potentially, cardiovascular health, irrespective of whether they are treated with ADT
Maréchal et al., 2018, [90]	Prostate and breast	69	12 weeks	Fourteen participants completed 12 weeks of a mixed exercise program (*n* = 6) or stretching (*n* = 8) while they were under cancer treatment	Exercise program led to significant improvements in physical capacity and may reduce sedentary behavior time
Park et al., 2012, [91]	Prostate	69.1	12 weeks	A total of 66 patients were randomized to an exercise or a control group. The exercise group received a combined exercise intervention (resistance, flexibility, and Kegel exercises) twice a week for 12 weeks, and the control group received only Kegel exercises	A 12-week combined exercise intervention after radical prostatectomy results in improvement of physical function, continence rate, and quality of life
Winters-Stone et al., 2015, [92]	Prostate	70	1 year	A total of 51 prostate cancer survivors were randomized to moderate to vigorous intensity resistance training or stretching (placebo)	One year of resistance training improved muscle strength in androgen-deprived prostate cancer survivors. Strengthening muscles using functional movement patterns may be an important feature of exercise programs designed to improve perceptions of physical function and disability
Pysizora et al., 2017 [93]	Advanced cancer	72	2 weeks	Sixty patients diagnosed with advanced cancer receiving palliative care were randomized into two groups: the treatment group (n = 30) and the control group (n = 30). The therapy took place three times a week for 2 weeks. The 30-min physiotherapy session included active exercises, myofascial release, and proprioceptive neuromuscular facilitation (PNF) techniques. The control group did not exercise	The physiotherapy program, which included active exercises, myofascial release, and proprioceptive neuromuscular facilitation techniques, had beneficial effects on cancer-related fatigue and other symptoms in patients with advanced cancer who received palliative care
Stuecher et al., 2019, [94]	Advanced gastrointestinal	67	12 weeks	Participants (n = 44) were randomly assigned to a home-based physical activity program of 150 min moderate walking per week or a control group (CG)	A home-based physical activity improves postural sway and body composition and might stabilize functional capacity in patients with advanced gastrointestinal cancer during chemotherapy. Although the other outcomes did not differ between groups, the aforementioned effects might contribute to the maintenance of independence in ADL and a better treatment tolerance and thus enhance patients’ quality of life

Initially, due to a poor representation of older patients in clinical trials, it was difficult to systematize the evidence in reviews of the literature [116]. In a recent meta-analysis of 1748 patients with prostate cancer (mean age > 69 years), exercise improved body composition with a significant positive overall effect in percent body fat (−1.0%, 95% CI = −1.3 to −0.6%), fat mass (−0.6 kg, 95% CI = −0.8 to −0.3 kg), trunk fat mass (−0.3 kg, 95% CI = −0.6 to −0.2 kg), lean mass (0.5 kg, 95% CI = 0.3 to 0.7 kg), and appendicular lean mass (0.4 kg, 95% CI = 0.2 to 0.6 kg). The effects on many parameters, considered fundamental indices of the individual’s functional capacity, are also important: time to per- form the 30-s sit-to-stand repetitions (2.8 reps, 95% CI = 1.7 to 4.0 reps), repeated sit-to -stand test (−1.0 s, 95% CI = −1.4 to −0.6 s), 400-m walk (−8.3 s, 95% CI = −12.4 to −4.2 s), 6-m fast walk (−0.1 s, 95% CI = −0.2 to −0.0), and stair climb (−0.2 s, 95% CI = −0.3 to −0.1 s). Of all the data presented in this meta-analysis, the one most frequently reported is that on the increase in VO_2_ at peak exercise, with an increase of 1.3 mL/kg·min in patients in the intervention group with exercise [117]. This parameter is very important because it is one of the most validated indices of cardiorespiratory fitness and functional capacity and its increase in different settings is often associated with a decrease in all-cause mortality [118,119]. Oncological disease in itself “ages” individuals: a reduction in cardiorespiratory fitness was observed in 50-year-old women with breast cancer of more than 30% compared to healthy controls. The mere fact of having an oncological disease, these patients “aged” by more than 10 years [120]. Again, physical exercise with the maintenance or even improvement of physical function has a positive pleiotropic effect. In a recent meta-analysis of 8109 patients with cancer of all types over the age of 60, improved physical function was significantly associated with lower all-cause mortality than patients with lower physical function [121].

The maintenance of functional abilities, therefore, becomes fundamental. From this point of view, the prevention of falls is one of the cornerstones of the prevention of disability. Balance training has been shown to be effective in improving chemotherapy-induced peripheral neuropathy in cancer patients over the age of 70 [122]. It would be important to include as many patients as possible in exercise trials because the patients who have early gait disturbances are the ones who could benefit most from exercise [123].

The impact of the diagnosis of a neoplastic pathology is a trigger that fuels mood disorders. A period of only 6 weeks of exercise in which aerobic and anaerobic activity was combined proved sufficient to improve depressive symptoms, especially in those who had worse symptoms at baseline. The median age of the patients was 68 years [80].

The beneficial effects of aerobic physical exercise are by now a consolidated fact with numerous evidence in the literature. Much remains to be verified regarding resistive type exercise and especially in elderly cancer patients where specific protocols are still lacking. However, the experience of rehabilitation cardiology with physical exercise tailored based on the patient’s stratification depending on the CPET is one of the validated approaches to guide clinicians to choose the type of exercise. The concept that even resistive exercise is effective and beneficial in the elderly patient and even in the elderly and sarcopenic patients is starting to make room in the literature [124,125,126]. In this sense, the prescription of resistive physical exercise could be even simpler for the clinician, perhaps by organizing the training protocols as real “pills” to be taken daily.

The outcomes to be pursued in elderly patients with cancer are still debated in the scientific community. Independent of age, the patient’s comprehensive assessment is the real litmus test of that individual’s performance status, both from a motor and cognitive point of view. Regardless of these considerations, physical exercise, accompanied or not by other multimodal rehabilitation inputs, has been shown to be effective in improving the quality of life of geriatric cancer patients [127,128]. The relevance of this evidence goes beyond the clinical setting alone, but they expand to the social field.

## 5. Prehabilitation: Exercise Training First 

The evidence of the efficacy of rehabilitation programs focused on physical exercise in geriatric cancer patients is consistent and is increasing from year to year. However, what happens before the patient starts cancer treatment, be it medical or surgical? In a meta-analysis of 10,030 geriatric cancer patients with different types of cancer, it was shown that higher cardiopulmonary test performance and higher peak VO_2_ values during exercise are significantly associated with better postoperative outcomes [129]. These data are of very high clinical importance and the cardiopulmonary test is rightly inserted as one of the best methods to stratify patients, both those referred to oncological rehabilitation but also for direct therapeutic decision-making. The main problem in geriatric cancer patients is not represented by overtreatment, but the undertreatment of patients mistakenly considered disabled when in fact they are frail. Cardiopulmonary exercise stress testing could provide objective measures to correctly stratify patients. With a view to better stratification of patients, the six-minute walk test also proved effective in “predicting” the number of complications based on the distance traveled. Patients with shorter walking distances were more likely to develop postoperative complications [130]. 

Considering this evidence, over the years prehabilitation programs have been born to better prepare patients for cancer treatments, both surgical and medical [131]. Based on some studies that have demonstrated the feasibility of a prehabilitation program in geriatric patients with cancer [132,133], numerous studies are underway, in different disease settings, with the aim of demonstrating the effectiveness of a rehabilitation program before starting anti-cancer treatment. One is a multi-center study of over 700 patients which aims to demonstrate the efficacy of multimodal prehabilitation in cancer patients undergoing elective surgery for colorectal cancer [134]. It should be noted that the clinical insignificance in a study of 418 elderly frail patients with a mean age of 78 years destined for colorectal resection is probably due to the scarce time devoted to physical exercise in the 4 weeks of the prehabilitation period [135].

## 6. Conclusions

Epidemiological trends for the healthcare system suggest that the aging population, frailty, and cancer are the newest challenges. Slowing the loss of homeostatic compensation is a critical point in avoiding a wave of disabled patients that healthcare systems are not ready to deal with. Even in geriatric cancer patients, the increase in physical activity and a correct nutritional intake are effective in improving their quality of life, a fundamental outcome in this population especially from a disability prevention perspective [136]. Population defrailing is one of the new frontiers of medicine, with enormous social and welfare implications. Recent data of a multicenter RCT suggest that a multicomponent intervention combining physical activity and nutrition is effective in preventing mobility-disability in frail elderly patients (average age of patients 78.9 years) [4]. Mortality and morbidity are no longer the main outcomes to be explored. Functional capacity, disability, and quality of life are becoming increasingly important in cancer patients undergoing CORE [137]. In this cohort, clinical trials are eagerly awaited, hopefully including elderly patients with cancer, multimorbidity, frailty, and disability to test new interventional models, programs, or more realistic functional outcomes.

## Figures and Tables

**Figure 1 cancers-15-01671-f001:**
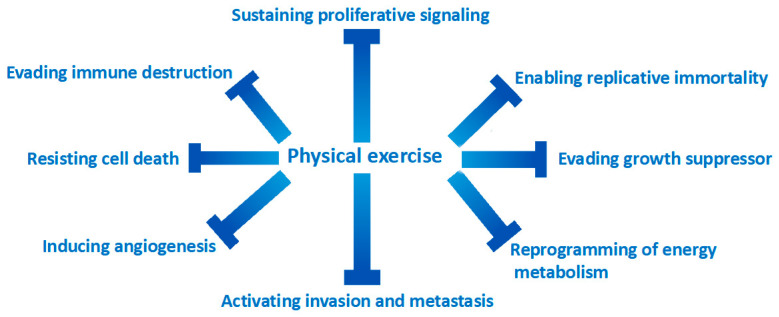
Exercise is effective in counteracting all the mechanisms of action responsible for the transformation of a normal cell into a cancer cell.

**Figure 2 cancers-15-01671-f002:**
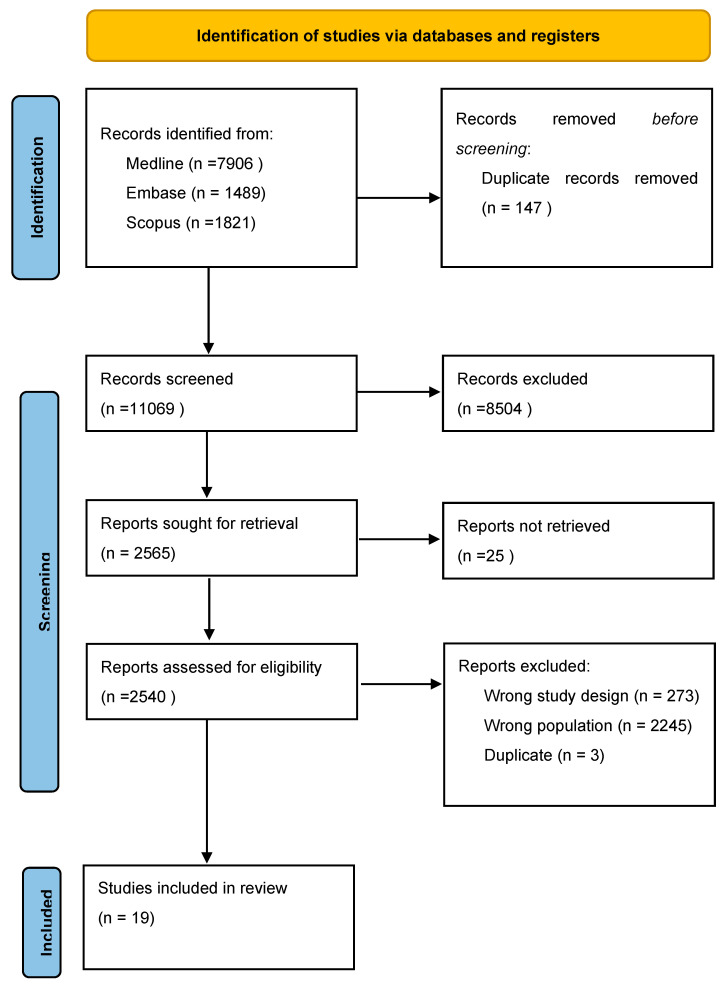
PRISMA diagram of selected studies.

**Figure 3 cancers-15-01671-f003:**
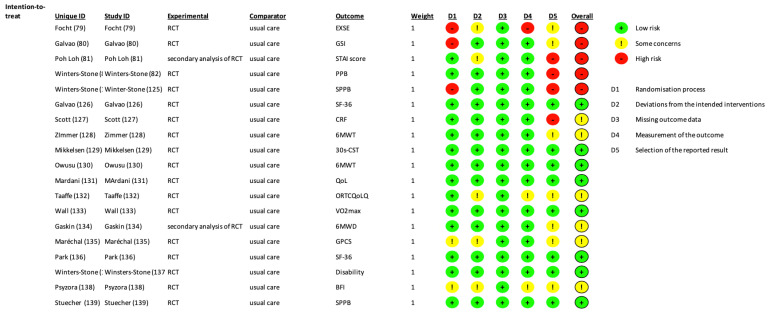
Risk of Bias of the included studies according to the ROB2 tool.

**Table 1 cancers-15-01671-t001:** Main characteristics and results of preclinic studies included in the systematic review.

Authors	Type of Study	Cancer Hallmark	Biological Mechanism Involved	Main Results
Xie L et al., 2007. [44]	Comparing the effects of weight loss by dietary calories restriction-fed or exercise-trained mice on Ras-MAPK and PI3K- Akt cascades	Sustaining proliferative signaling	Impact on Plasma IGF-1 and skin tissue IGF-1R levels -Impact on Ras-MAPK Pathway -Impact on PI3K Pathway -Effects on Plasma leptin levels -Effect on activated caspase-3 levels in skin tissues -Effects of gene expression relevant to RAS and -PI3K signaling pathways	Body weight significantly decreased, with a 20% dietary calories restriction, which corresponded to a decrease of body fat and plasma IGF-1 levels. Treadmill exercise with pair-feeding selectively reduced the PI3K-Akt pathway
Ouyang P et al., 2010. [45]	Impact of exercise with dietary consideration on the phospholipid profile in TPA-induced mouse skin tissues	Sustaining proliferative signaling	Impact on PI3K Pathway	Exercise with controlled diet interventions significantly reduced body weight and body fat as well as modified the phospholipid profile. This modified profile might provide potential cancer prevention benefits, perhaps via reducing TPA-induced PIs and PI-related PI3K expression.
Leung PS et al., 2004. [46]	Effect of exercise on serum-stimulated p53 protein content in the LNCaP prostate cell line.	Sustaining proliferative signaling	Effects on circulating IGF-1 levels and p53 expression	Serum from men who regularly exercise does increase the p53 protein in LNCaP tumor cells and that this is likely important for the reduced cell growth and the induction of apoptosis.
Yu M et al., 2016. [47]	Female SENCAR mice were pair-fed an AIN-93 diet with or without 10-week treadmill exercise at 20 m/min, 60 min/day and 5 days/week.	Evading growth suppressor	Effects of exercise on MDM2 and p53 expression Effects of exercise on the expression of p53-transcripted proteins (p21, IGFBP-3, PTEN)	Exercise appeared to activate p53, resulting in enhanced expression of p21, IGFBP-3, and PTEN that might induce a negative regulation of IGF-1 pathway and thus contribute to the observed cancer prevention by exercise in this skin cancer model
Piguet AC et al., 2015. [48]	Mice were fed a standardized 10% fat diet and were randomly divided into exercise or sedentary groups. The exercise group ran on a motorized treadmill for 60 min/day, 5 days/week for 32 weeks	Evading growth suppressor	Exercise can stimulate the phospshorylation of AMPK and its substrate raptor, which decrease the kinase activity of mTOR	After 32 weeks of regular exercise, 71% of exercised mice developed nodules larger than 15 mm^3^ vs. 100% of mice in the sedentary group. The mean number of tumors per liver was reduced by exercise, as well as the total tumoral volume per liver
Jiang Wet al, 2009. [49]	Effects on mammary carcinogenesis of physical activity in rats.	Evading growth suppressor	Effects of exercise on caspase-3 activity and AMPK signaling	Cell proliferation associated proteins were reduced and caspase 3 activity and pro-apoptotic proteins were elevated by PA or RE relative to SC (*p* < 0.05). It was observed that these effects may be mediated, in part, by activation of AMP-activated protein kinase and down regulation of protein kinase B and the mammalian target of rapamycin.
Zhu Z el al., 2008. [50]	Identify circulating growth factors, hormones, and cellular and molecular mechanisms that account for the effects of physical activity on mammary carcinogenesis in rats.	Evading growth suppressor	Effects of physical exercise on activation of AMP-activated protein kinase, and down-regulation of protein kinase B, which collectively down-regulate the activity of the mammalian target of rapamycin	Cancer incidence (98.1 versus 84.6%; *p* < 0.01) and average number of cancers per rat (3.72 versus 2.67, respectively; *p* < 0.01) were reduced by physical activity. The average cancer mass per rat was 0.62 g in the physically active group and 1.16 g in the sedentary control group (*p* = 0.17)
Zhu Z el al., 2008. [50]	Identify circulating growth factors, hormones, and cellular and molecular mechanisms that account for the effects of physical activity on mammary carcinogenesis in rats.	Evading growth suppressor	Effects of physical exercise on activation of AMP-activated protein kinase, and down-regulation of protein kinase B, which collectively down-regulate the activity of the mammalian target of rapamycin	Cancer incidence (98.1 versus 84.6%; *p* < 0.01) and average number of cancers per rat (3.72 versus 2.67, respectively; *p* < 0.01) were reduced by physical activity. The average cancer mass per rat was 0.62 g in the physically active group and 1.16 g in the sedentary control group (*p* = 0.17)
Khori V et al., 2015. [51]	Forty-eight female BALB/c mice were equally divided into six groups to investigate the effects of interval exercise training with tamoxifen on miR-21 expression and its possible assumed mechanisms in an estrogen receptor-positive breast cancer model	Evading growth suppressor	Exercise training and tamoxifen reduced tumor IL-6 levels, NF-kB and STAT3 expressions, and up-regulated TPM1 and PDCD4 expressions (*p* < 0.05). Both exercise and tamoxifen had synergistic effects in reducing miR-21 and Bcl-2, and up-regulating PDCD4 expression	Results showed that interval exercise training may reduce mammary tumor burden in mice through possible underlying pathway of miR-21
Zheng X et al., 2008. [52]	Effect of voluntary exercise on the formation and growth of human pancreas Panc-1 and prostate PC-3 tumors in immunodeficient mice	Resisting cell death	Effects of voluntary running wheel exercise on mitosis and apoptosis in Panc-1 and PC-3 tumors	Voluntary running wheel exercise inhibited the growth of human pancreas and prostate tumors in immunodeficient SCID mice, and these effects of exercise were paralleled by decreased proliferation and increased apoptosis
Hojman P et al., 2011. [53]	Incubation of mamary cancer cells with conditioned serum from exercising mice	Resisting cell death	Effect of exercise on caspase activation	Post exercise serum inhibits mammary cancer cell proliferation and induces apoptosis
Barnard RJ et al., 2007. [54]	Serum from sedentary controls or men with regular (5 days/week) aerobic exercise was used to stimulate lymph node cancer of the prostate (LNCaP) tumor cells in vitro	Resisting cell death	Effect of exercise on the activity of p53, p21 and Bcl-2	Exercise training alters serum insulin-like growth factor axis factors in vivo that increase LNCaP cellular p53 protein content in vitro leading to reduced growth via p21 and induced apoptosis via the mitochondrial pathway
Higgins KA et al., 2014. [55]	Luciferase-tagged A549 lung adenocarcinoma cells were injected through the tail vein of nude male mice. After lung tumors were identified, the mice were randomized to daily wheel running versus no wheel running	Resisting cell death	Effect of exercise on p53 levels and mediators of apoptosis including Bax and active caspase 3	Lung tumors in exercising mice grew significantly more slowly relative to sedentary mice
Betof AS et al., 2015. [56]	Estrogen receptor–negative (ER-, 4T1) and ER+ (E0771) tumor cells were implanted orthotopically into syngeneic mice (BALB/c, N = 11–12 per group) randomly assigned to exercise or sedentary control	Resisting cell death	Effect of exercise on apoptosis	Exercise plus chemotherapy prolonged growth delay compared with chemotherapy alone (*p* < 0.001) in the orthotopic 4T1 model (n = 17 per group). Exercise is a potential novel adjuvant treatment of breast cancer
Jones LW et al., 2010. [57]	Athymic female mice fed a high-fat diet were orthotopically (direct into the mammary fat pad) implanted with human breast cancer cells into the right dorsal mammary fat pad and randomly assigned [1:1] to voluntary wheel running (*n* = 25) or a nonintervention (sedentary) control group (*n* = 25).	Inducing angiogenesis	Activity of VEGF and hypoxia-inducible factor (HIF-1) Activity of AMPK and peroxisome proliferator-activated receptor-γ coactivator (PGC)-1α	Aerobic exercise can significantly increase intratumoral vascularization, leading to “normalization” of the tissue microenvironment in human breast tumors
Jones LW et al., 2012. [58]	C57BL/6 male mice (6–8 wk of age) were orthotopically injected with transgenic adenocarcinoma of mouse prostate C-1 cells (5 × 10^5^) and randomly assigned to exercise (*n* = 28) or a non-intervention control (*n* = 31) group	Inducing angiogenesis	Effects of primary tumor growth and metastasis; -Effects of prometastatic gene expression; -Effects on tumor MAPK and PI3K signaling; Effects on HIF-1, metabolism, and angiogenesis; -Effects on tumor perfusion/diffusion and vessel function and maturation	Exercise promote stabilization of HIF-1α with subsequent upregulation of a proangiogenic phenotype stimulating “productive” tumor perfusion (vascularization) with a shift toward reduced metastasis in an orthotopic model of murine prostate cancer
McCullough DJ et al., 2014. [59]	Prostate tumor blood flow, vascular resistance, patent vessel number, and hypoxia were measured in vivo in conscious rats at rest and during treadmill exercise, and vasoconstrictor responsiveness of resistance arterioles was investigated in vitro	Inducing angiogenesis	Effect of exercise on tumor blood flow	During exercise there is enhanced tumor perfusion and diminished tumor hypoxia due, in part, to a diminished vasoconstriction. The clinical relevance of these findings is that exercise may enhance the delivery of tumor-targeting drugs as well as attenuate the hypoxic microenvironment within a tumor and lead to a less aggressive phenotype
Almeida PW et al., 2009. [60]	Effect of training protocol in Swiss Mice inoculated with Ehrlich tumor cells	Evading immune destruction	Accumulation in tumor tissue of immunocompetent cells	Moderate swim training markedly reduced the growth of Ehrlich tumors in mice and suppressed macrophage infiltration and neutrophil accumulation in tumor tissue
Zielinski MR et al., 2004. [61]	Female BALB/c mice were randomly assigned to sedentary control or daily exercised groups	Evading immune destruction	-Effects of exhaustive exercise on EL-4 tumor progression and regression; -Exercise-induced changes in intratumoral cellular composition; -Effects of exercise on markers of cell proliferation and death	Intense exercise influences the microenvironment of a subcutaneously transplanted allogeneic tumor. Daily intense, prolonged exercise caused a delay in tumor growth, a decrease in the number of inflammatory cells (macrophages and neutrophils), and a decrease in the number of blood vessels within the tumors
Baltgalvis KA, et al., 2008. [62]	The effect of exercise on biological pathways in *Apc ^Min^* mouse intestinal polyps	Evading growth suppressor	Effect of exercise on apoptosis, b*-Catenin,* growth signaling,	Exercise can regulate *Apc^Min^* mouse intestinal polyp composition
Abdalla DR et al., 2013. [63]	Cytokine synthesis by lymphocytes in the presence of mammary tumors and the interaction with physical activity	Evading immune destruction	Assessing of cluster of differentiation (CD)3, CD4, and CD8 markers and the expression of interferon-γ, interleukin (IL)-2, IL-4, IL-10, IL-12, transforming growth factor β, and tumor necrosis factor α cytokines	Physical activity promoted reductions in the incidence of tumor development and promoted immune system polarization toward an antitumor Th1 response pattern profile
Abdalla DR et al., 2014. [64]	Effect of exercise on female BALB/c virgin mice	Evading immune destruction	Effect of exercise on effector of immunity	Practicing physical activity in the presence of a tumor promoted a reduction in tumor development and polarized the immunological response in the direction of the antitumor M1 profile
Pedersen L et al., 2016. [65]	Effect of exercise on tumor-bearing mice	Evading immune destruction	Training-Dependent Reduction in Tumor Growth Is Associated with Induction of Immune-Related Pathways; -Training Regulates Tumor Growth through Intratumoral NK Cell Infiltration	Voluntary wheel running inhibits tumor onset and progression across a range of tumor models and anatomical locations
Glasner A et al., 2012. [66]	All experiments were performed using 6- to 8-wk-old mice of the C57BL/6 background	Evading immune destruction	B16 and D122 cells express a ligand or ligands for NKp46/NCR1; NKp46/NCR1-dependent NK degranulation following incubation with B16 and D122	Enhancing NKp46/NCR1 activity either through the elevation of its expression or by cytokine activation might be beneficial for the treatment of tumor metastasis
Hampras SS et al., 2012. [68]	Epidemiological study to evaluate predictors of Treg levels in a cohort of healthy women	Evading immune destruction	Effect of a wide range of factors on Treg function	Exercise (3 or more days/week) was found be significant negative predictors of Treg cell levels
Aoi W et al., 2013. [70]	DNA microarrays were used to compare the transcriptome of muscle tissue in sedentary and exercised young and old mice	Myokines and cancer	Effect of exercise on level of circulating secreted protein acidic and rich in cysteine (SPARC)	Exercise stimulates SPARC secretion from muscle tissues and that SPARC inhibits colon tumorigenesis by increasing apoptosis

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
