# Peer review of "Exercise Training in Elderly Cancer Patients: A Systematic Review"

_cancers, 2023, doi:10.3390/cancers15061671_

Round 1

Reviewer 1 Report (Previous Reviewer 1)

Thank you to the authors for the revised manuscript. I still believe this paper would help provide a helpful summary of the evidence in the literature supporting exercise training for older cancer survivors.

The authors have described their paper as a systematic review in this revision, but it's not clear that the review was truly systematic. For example, the authors need to describe the date range for the articles they reviewed (and why they selected that date range), and the criteria (in terms of study samples, study designs, etc.) that they limited their search to. They have described including articles that include patients older than 65 years, but there are likely hundreds of additional articles published that include patients who are older than 65 years. The authors might decide on (and cite) a definition of "elderly cancer survivor" and limit their review to articles that included only that age range. As written, they have reviewed articles with average participant ages of 63-77 years, but those may have also included younger cancer survivors.

The article also reads a bit disjointed, particularly the transition in focuses from pathophysiological mechanisms of exercise in elderly cancer survivors to the results of clinical trials. The authors describe table 1 as including the descriptions of all of the articles they reviewed, but this seems to be only covering the pathophysiological mechanism articles they reviewed.

The authors might have an easier time framing this article as a "narrative review" in which they're highlighting key pieces of evidence regarding exercise training and older cancer survivors, including recommendations for clinical practice, future research targets, etc.

Author Response

Reviewer 1

Thank you to the authors for the revised manuscript. I still believe this paper would help provide a helpful summary of the evidence in the literature supporting exercise training for older cancer survivors. The authors have described their paper as a systematic review in this revision, but it's not clear that the review was truly systematic.                                                                                                 For example, the authors need to describe the date range for the articles they reviewed (and why they selected that date range), and the criteria (in terms of study samples, study designs, etc.) that they limited their search to.                                                                                                                           They have described including articles that include patients older than 65 years, but there are likely hundreds of additional articles published that include patients who are older than 65 years.The authors might decide on (and cite) a definition of "elderly cancer survivor" and limit their review to articles that included only that age range.As written, they have reviewed articles with average participant ages of 63-77 years, but those may have also included younger cancer survivors.

We are thankful to Reviewer for Her/His suggestions. We have extensively revised section 4 of our article to make the thread of discussion more linear in order to fill the gap between the previous section in which the effectiveness of physical exercise in preclinical settings is analyzed, and section 4 where we try to analyze what happens in a clinical setting. Going to systematically analyze this population in the indicated setting is very difficult, both because the scientific community has only recently begun to consider physical exercise as a real medication but above all because the elderly patient is generally excluded from randomized controlled trials due to dynamics that in no way reflect real life. One of the reasons why elderly patients are not included in these trials is precisely because the social and healthcare network in which they are inserted does not allow their recruitment. Our article attempts to bring to light new therapeutic approaches for the elderly oncological patient by radically changing the objectives of oncological care in the elderly patient: no longer strictly focused on the oncological disease but rather on the individual as a whole. Such complexity is difficult to frame in the dichotomy of a randomized controlled trial but this work attempts to convince the scientific community to invest in this direction. As regards the definition of long-term cancer survivor, this is borrowed from Bluethemann and colleagues who, in their work "Anticipating the Silver tsunami: Prevalence and Trajectories and Comorbidity Burden among Older Cancer Survivors in the United States" depict the epidemiological trajectory of oncological diseases in the near future.                                                                                                                                       Although the definition takes into consideration patients who have reached the age of 65, from a biological point of view, including patients who are slightly younger is functional for our goal: to try to convince the scientific community that a geriatric approach to the patient is effective and above all sustainable in the prevention of disability.

The article also reads a bit disjointed, particularly the transition in focuses from pathophysiological mechanisms of exercise in elderly cancer survivors to the results of clinical trials.                                                                                                                             Thanks for your suggestion. Section 4 has been modified to smooth the transition from the molecular analysis of the benefit of exercise in preclinical setting to the review of articles in clinical settings where this benefit is demonstrated.

The authors describe table 1 as including the descriptions of all of the articles they reviewed, but this seems to be only covering the pathophysiological mechanism articles they reviewed.    Thanks for the tip, that was a typo: Table 1 summarizes the preclinical studies mentioned in the previous section.

The authors might have an easier time framing this article as a "narrative review" in which they're highlighting key pieces of evidence regarding exercise training and older cancer survivors, including recommendations for clinical practice, future research targets, etc.               

As already highlighted in the first review phase, we initially planned to publish a narrative review of the literature, but the meticulous and in-depth preparatory work for this article provided the necessary foundations to be able to pursue a systematic review work also according to reviewer 2 suggestions.

Reviewer 2 Report (Previous Reviewer 2)

Dear authors,

It is obvious that by implementing PRISMA guidelines during the literature research you invested time and energy revising this manuscript. However, PRISMA guidelines are not limited to depicting the literature research process and a flow diagram and therefore your article is currently something in between a narrative and a systematic review. What I am specifically missing for a systematic review is the risk of bias (RoB) assessment. Since you only included 9 articles, it won't be too much additional work but a lot of additional information for the readers.

If you might lack experience in this regard, I can recommend the respective RoB assessment tools from the Cochrane Collaboration for RCTs (RoB 2) and non-randomized trials (ROBINS-I) to you. They are of high quality and easy to use. 

If you included a RoB assessment with a respective table, then I would consider accepting your article as a systematic review. 

Author Response

Reviewer 2

Dear authors,                                                                                                                                    It is obvious that by implementing PRISMA guidelines during the literature research you invested time and energy revising this manuscript. However, PRISMA guidelines are not limited to depicting the literature research process and a flow diagram and therefore your article is currently something in between a narrative and a systematic review.                                                                                            What I am specifically missing for a systematic review is the risk of bias (RoB) assessment. Since you only included 9 articles, it won't be too much additional work but a lot of additional information for the readers. If you might lack experience in this regard, I can recommend the respective RoB assessment tools from the Cochrane Collaboration for RCTs (RoB 2) and non-randomized trials (ROBINS-I) to you. They are of high quality and easy to use. If you included a RoB assessment with a respective table, then I would consider accepting your article as a systematic review.                                                                                             

As requested, we have implemented a figure that evaluates the risk of bias and we have used the ROB2 tool. We thank you for the suggestion.

Reviewer 3 Report (Previous Reviewer 3)

The authors significantly improved the quality of the manuscript.

Author Response

The authors significantly improved the quality of the manuscript.                             Thank you so much for you revision.                                                                        

Round 2

Reviewer 1 Report (Previous Reviewer 1)

Thanks for the opportunity to review the revised version of this manuscript. I think the authors still need to strengthen the criteria by which they define their systematic review, more notably for the clinical trials they have reviewed and presented in Table 2. Otherwise, I would still contend that this is more of a narrative review than a systematic review.

The authors have clarified that they have limited their search to the last decade, stating that "the prescription of physical exercise as medicine is an acquisition of the last decade (124), for this reason 269 we have included only clinical trials that went in this direction." The article the authors have cited to justify this is a position statement about cardiac rehabilitation published in 2012. The first American College of Sports Medicine Roundtable on Exercise Guidelines for Cancer Survivors (Schmitz et al., MSSE, 2010) was published 2 years before this cardiac rehabilitation position statement, and it included evidence from articles including older cancer survivors. There were numerous randomized controlled trials involving exercise and including older cancer survivors published before 2012. I do not think the authors have provided enough justification to limit their review to 2012-2022; they may instead consider reframing the article as a narrative review providing an update on more recent evidence.

Furthermore, the authors have not sufficiently described the inclusion/exclusion of articles in the group they reviewed. They include Winters-Stone (2012), with mean participant age 63 and range 50-83. It's not clearly outlined, but I am assuming they are limiting to articles with at least mean age of 60. There are a number of additional randomized controlled trials of exercise for cancer survivors published between 2012-2022 that also had mean ages of 60+. Examples include Wiskemann et al., 2019 (in Pancreas), which includes patients with mean age 60 years; Cheville et al., 2013 (J of Pain and Symptom Management), which includes patients with mean age ~63-65; Gillis et al., 2014 (Anesthesiology). There are many more that show up in quick searches in PubMed and/or Google Scholar.

Given these issues, I content that the authors either need to (1) expand their review (and truly be systematic) to include all relevant articles, (2) more clearly define their inclusion/exclusion criteria to show that they have already performed a systematic review, or (3) reframe their article as a narrative review.

Author Response

Dear authors,

Thanks for the opportunity to review the revised version of this manuscript. I think the authors still need to strengthen the criteria by which they define their systematic review, more notably for the clinical trials they have reviewed and presented in Table 2. Otherwise, I would still contend that this is more of a narrative review than a systematic review.

We thank the reviewer for his/her suggestions. We have extensively edited the body of the manuscript, and have completely revised the selection of articles. We selected 19 randomized controlled trials that either recruited elderly cancer patients or whose mean age of recruited patients was greater than or equal to 65 years.We also pointed to a trial in which the older cancer survivor was mentioned (125) .

The authors have clarified that they have limited their search to the last decade, stating that "the prescription of physical exercise as medicine is an acquisition of the last decade (124), for this reason 269 we have included only clinical trials that went in this direction." The article the authors have cited to justify this is a position statement about cardiac rehabilitation published in 2012. The first American College of Sports Medicine Roundtable on Exercise Guidelines for Cancer Survivors (Schmitz et al., MSSE, 2010) was published 2 years before this cardiac rehabilitation position statement, and it included evidence from articles including older cancer survivors. There were numerous randomized controlled trials involving exercise and including older cancer survivors published before 2012. I do not think the authors have provided enough justification to limit their review to 2012-2022; they may instead consider reframing the article as a narrative review providing an update on more recent evidence.

As noted above, with thanks to the reviewer for the suggestion, we have completely changed the selection of studies.

Furthermore, the authors have not sufficiently described the inclusion/exclusion of articles in the group they reviewed. They include Winters-Stone (2012), with mean participant age 63 and range 50-83. It's not clearly outlined, but I am assuming they are limiting to articles with at least mean age of 60. There are a number of additional randomized controlled trials of exercise for cancer survivors published between 2012-2022 that also had mean ages of 60+. Examples include Wiskemann et al., 2019 (in Pancreas), which includes patients with mean age 60 years; Cheville et al., 2013 (J of Pain and Symptom Management), which includes patients with mean age ~63-65; Gillis et al., 2014 (Anesthesiology). There are many more that show up in quick searches in PubMed and/or Google Scholar.

Study selection has been completely overhauled. The new studies are summarized in the reference table (Table 2) and the selection took place according to the PRSMA criteria. Figure three also shows the assessment of the risk of bias using the RoB2 tool as previously indicated.

Given these issues, I content that the authors either need to (1) expand their review (and truly be systematic) to include all relevant articles, (2) more clearly define their inclusion/exclusion criteria to show that they have already performed a systematic review, or (3) reframe their article as a narrative review.

The revised manuscript has been rebuild as a systematic review including all relevant articles. Inclusion/exclusion criteria have been specified to perform systematic review and the article has been reframed as a narrative review.

Round 3

Reviewer 1 Report (Previous Reviewer 1)

Thanks for the opportunity to review this manuscript revision. The authors have improved their methods and descriptions of the articles they reviewed, making this a more rigorous systematic review.

Author Response

We are thankful to the Reviewer for Her/His comments. We are delighted that the revised version has been appreciated by the Reviewer.

This manuscript is a resubmission of an earlier submission. The following is a list of the peer review reports and author responses from that submission.

Round 1

Reviewer 1 Report

This review does a nice job summarizing the evidence supporting exercise interventions for older cancer patients and survivors. Figure 1 is a nice graphical representation of the pathophysiological basis of exercise in older adults, and Table 1 provides a thorough review of the evidence of related benefits from (mostly) preclinical trials. The addition of a similar figure and table to summarize the evidence from clinical trials described in sections 4 and 5 would help complete and balance the paper.

Reviewer 2 Report

Dear authors,

Your manuscript "EXERCISE TRAINING IN ELDERLY CANCER SURVIVOR" treats an important and relevant topic. It reads very well and I am convinced that it will be interesting for many readers of Cancers.

However, in my opinion there are serious problems with your methodology. Since this is no systematic review, you don't implement the very reasonable PRISMA-technique. There is no clear research question or hypothesis, no inclusion- or exclusion criteria for the literature you analyzed and discussed, no risk of bias assessment, etc. 

So in my opinion, in the current form the manuscript would be a very nice book chapter but not a research article as it lacks the necessary methodology. 

Reviewer 3 Report

Thank you for the opportunity to review this comprehensive article on exercise training in elderly cancer survivors. Below are a few points that need to be addressed before the article is considered for publication.

-In the title, the authors consider adding the word „patients“ and survivors for better clarity since the manuscript contains the prehabilitation phase.

-Why have authors focused on training elderly survivors? What makes them different from other rehabilitation interventions? Please address this point

-Is a type of exercise training intervention applicable to what types of cancer? At what stage of disease? Is training compatible with cancer treatment? Explain this issue

-Figure 1. Add a title for the figure. Furthermore, clarify whether it is the authors´ own or derived from the studies included. Please, clarify this issue

-4. Exercise in the older cancer survivor. Should be an exercise type „standard“ established at the end of the chapter based on included studies?

-Line 276 „The SARS-CoV-2 pandemic“ For a better overview, consider adding a short discussion on home-based exercise intervention in cancer survivors. These alternatives were used instead of a centre-based approach to overcome quarantine. 

-Moreover, the authors should discuss if the home-based approach is usable and safe for elderly cancer patients and survivors.

-Line 281, to support this vital statement on telerehabilitation, consider strengthening this statement with recent evidence on the rationale for integrating CORE and Telehealth. For this latter point, see Batalik L, et al. Cardio-Oncology Rehabilitation and Telehealth: Rationale for Future Integration in Supportive Care of Cancer Survivors. Front Cardiovasc Med. 2022;9:858334. Published 2022 Apr 15. doi:10.3389/fcvm.2022.858334

-The authors might consider including future perspectives for this population. As alternatives and telehealth have already been mentioned, do the authors think there is a future for using artificial intelligence for exercise training in elderly cancer survivors?